# Hippo signaling impairs alveolar epithelial regeneration in pulmonary fibrosis

**Rachel Warren†, Handeng Lyu†, Kylie Klinkhammer, Stijn P De Langhe\***

Department of Medicine, Division of Pulmonary and Critical Medicine, Mayo Clinic, Rochester, United States

**Abstract** Idiopathic pulmonary fibrosis (IPF) consists of fibrotic alveolar remodeling and progressive loss of pulmonary function. Genetic and experimental evidence indicates that chronic alveolar injury and failure to properly repair the respiratory epithelium are intrinsic to IPF pathogenesis. Loss of alveolar type 2 (AT2) stem cells or mutations that either impair their self-renewal and/or impair their differentiation into AT1 cells can serve as a trigger of pulmonary fibrosis. Recent reports indicate increased YAP activity in respiratory epithelial cells in IPF lungs. Individual IPF epithelial cells with aberrant YAP activation in bronchiolized regions frequently co-express AT1, AT2, conducting airway selective markers and even mesenchymal or EMT markers, demonstrating 'indeterminate' states of differentiation and suggesting that aberrant YAP signaling might promote pulmonary fibrosis. Yet, Yap and Taz have recently also been shown to be important for AT1 cell maintenance and alveolar epithelial regeneration after *Streptococcus pneumoniae*-induced injury. To investigate how epithelial Yap/Taz might promote pulmonary fibrosis or drive alveolar epithelial regeneration, we inactivated the Hippo pathway in AT2 stem cells resulting in increased nuclear Yap/Taz, and found that this promotes their alveolar regenerative capacity and reduces pulmonary fibrosis following bleomycin injury by pushing them along the AT1 cell lineage. Vice versa, inactivation of both *Yap1* and *Wwtr1* (encoding Taz) or *Wwtr1* alone in AT2 cell stem cells impaired alveolar epithelial regeneration and resulted in increased pulmonary fibrosis upon bleomycin injury. Interestingly, the inactivation of only *Yap1* in AT2 stem cells promoted alveolar epithelial regeneration and reduced pulmonary fibrosis. Together, these data suggest that epithelial Yap promotes, and epithelial Taz reduces pulmonary fibrosis suggesting that targeting Yap but not Taz-mediated transcription might help promote AT1 cell regeneration and treat pulmonary fibrosis.

**\*For correspondence:**
delanghe.stijn@mayo.edu

†These authors contributed equally to this work

**Competing interest:** The authors declare that no competing interests exist.

## Editor's evaluation

This is an interesting and potentially significant study that adds important new information to our understanding of the mechanisms of lung epithelial repair after tissue injury. The authors have delineated a novel and non-redundant role for the hippo pathway and the downstream regulators Yap/Taz in regulating the repair of lung injury. These studies will inform future investigations into the mechanisms of the repair of lung injury.

## Introduction

Idiopathic pulmonary fibrosis (IPF) pathogenesis encompasses alveolar and fibrotic remodeling, inflammation, and eventual loss of lung architecture *Barkauskas and Noble, 2014* resulting in progressive loss of pulmonary function, respiratory failure, and death, often within 5 years of diagnosis *King et al., 2011*; *Steele and Schwartz, 2013*. It is now largely accepted that key mechanisms of IPF initiation and progression include chronic alveolar injury and failure to properly repair the respiratory epithelium by AT2 cells *Camelo et al., 2014*; *Thannickal et al., 2004*; *Yang et al., 2013*. Histologically,

IPF is typified by respiratory epithelial cells in the lung parenchyma that express atypical proximal airway epithelial cell type markers including those expressed by goblet cells and BCs that are normally restricted to conducting airways and indeterminate cell type markers *Plantier et al., 2011*; *Seibold et al., 2013*. These atypical epithelial cells form bronchiolized or honeycomb structures that replace normal alveolar structures. Dramatic changes in ciliated, basal, and goblet-cell gene signatures correspond with the loss of genes associated with normal alveolar epithelial cells, reflecting profound changes in epithelial cells in both the airways and alveoli in IPF lungs *Seibold et al., 2013*; *Kropski et al., 2015*; *Xu et al., 2016*.

Yap/Taz signaling are important in alveolar epithelial regeneration by AT2 cells after *Streptococcus pneumoniae*-induced injury *LaCanna et al., 2019*. In this model, simultaneous inactivation of *Yap1* and *Wwtr1* (Taz) resulted in impaired alveolar epithelial regeneration, whereas single *Yap1* or *Wwtr1* mutant mice had minor or no phenotypes *LaCanna et al., 2019*. Interestingly, loss of *Yap1/Wwtr1* in AT1 cells leads to spontaneous reprogramming into the AT2 cell lineage. However, while overexpression of a dominant active *Yap1$^{S112A}$* in AT2 cells induced *Hopx* expression, these cells did not exhibit the classic flattened and elongated morphology of mature AT1 cells, and most continued to express *Sftpc Penkala et al., 2021*. *Yap1$^{S112A}$* overexpression often does not recapitulate the effects of Hippo pathway inactivation as *Yap1$^{S112A}$* induces a feedback activation of the Hippo pathway, resulting in increased degradation of endogenous Yap and Taz *Chen et al., 2015*; *Moroishi et al., 2015*. In addition, the inactivation of *Yap1* alone in AT1 cells has no effect on AT1 cell marker expression. These data suggest that nuclear Yap can drive the expression of some AT1 cell genes but may not be sufficient to fully reprogram AT2 cells into AT1 cells. However, the inactivation of Hippo kinases *Stk3/4* (Mst2/1, respectively), resulting in increased nuclear Yap and Taz, promotes the proliferation of AT2s and their differentiation into AT1s *Gokey et al., 2021*; *Little et al., 2021*. Interestingly, the inactivation of *Wwtr1* in adult mouse AT2 cells has been shown to result in reduced AT1 cell differentiation in organoids and bleomycin-induced lung injury *Sun et al., 2019*. Together these findings again suggest an important role for Taz in AT1 cell differentiation. Yet, in a pneumonectomy model inactivation of *Yap1* in AT2 cells (*Sftpc$^{CreER}$*; *Yap1$^{f/-}$*) decreased their proliferation and differentiation into AT1 cells compared to AT2 cells heterozygous for *Yap1* (*Sftpc$^{CreER}$*; *Yap1$^{f/+}$*) *Liu et al., 2016*, though comparison to wildtype (*Sftpc$^{CreER}$*; *Yap1$^{+/+}$*) AT2 cells was not performed.

Clearly, the Hippo pathway plays a very important role in lung development, homeostasis, and repair after injury. Although most of the available literature investigating the role of the Hippo pathway in the lung considers Yap and Taz as functionally redundant, some of the conflicting findings described above can only be explained by the fact that Yap and Taz play distinctive or even contrasting roles in different contexts and have dose-dependent as well as nuclear and cytoplasmic roles. Yap and Taz have common and distinctive structural features, reflecting only partially overlapping regulatory mechanisms and control nonidentical transcriptional programs. The issue of their divergent roles is currently underexplored in lung biology but holds fundamental implications for mechanistic and translational studies.

Many labs have found that AT2 to AT1 cell differentiation is impaired in IPF and that cells get stuck in a transitional state *Jiang et al., 2020*; *Kobayashi et al., 2020*. After lung injury, damage-associated transient progenitors (DATPs) emerge, representing a transitional state between AT2 stem cells and newly regenerated AT1 cells (aka pre-alveolar type-1 transitional cell state (PATS)) *Kobayashi et al., 2020*; *Choi et al., 2020*. Interestingly, DATP or PATS express profibrotic genes and show an enrichment of TP53, YAP, TGFβ, DNA-damage-response signaling, and cellular senescence, all of which are known to be involved in fibrosis in multiple organs, including the lung. However, the molecular pathways that induce and/or maintain DATPs are not completely understood nor is it clear whether promoting AT1 cell differentiation is sufficient to drive the resolution of pulmonary fibrosis. Here, by using murine lineage tracing coupled with injury-repair models and spatial transcriptomics studies, we uncovered a method to treat lung fibrosis by promoting AT1 cell differentiation through inhibition of *Yap1*. Mechanistically, the use of loss of function and spatial transcriptomics revealed an inhibition of *Wwtr1* and AT1 cell differentiation by Yap. Importantly, these transitional states correlate with abnormal epithelial cells associated with defective fibrotic foci in human lungs with progressive pulmonary fibrosis.

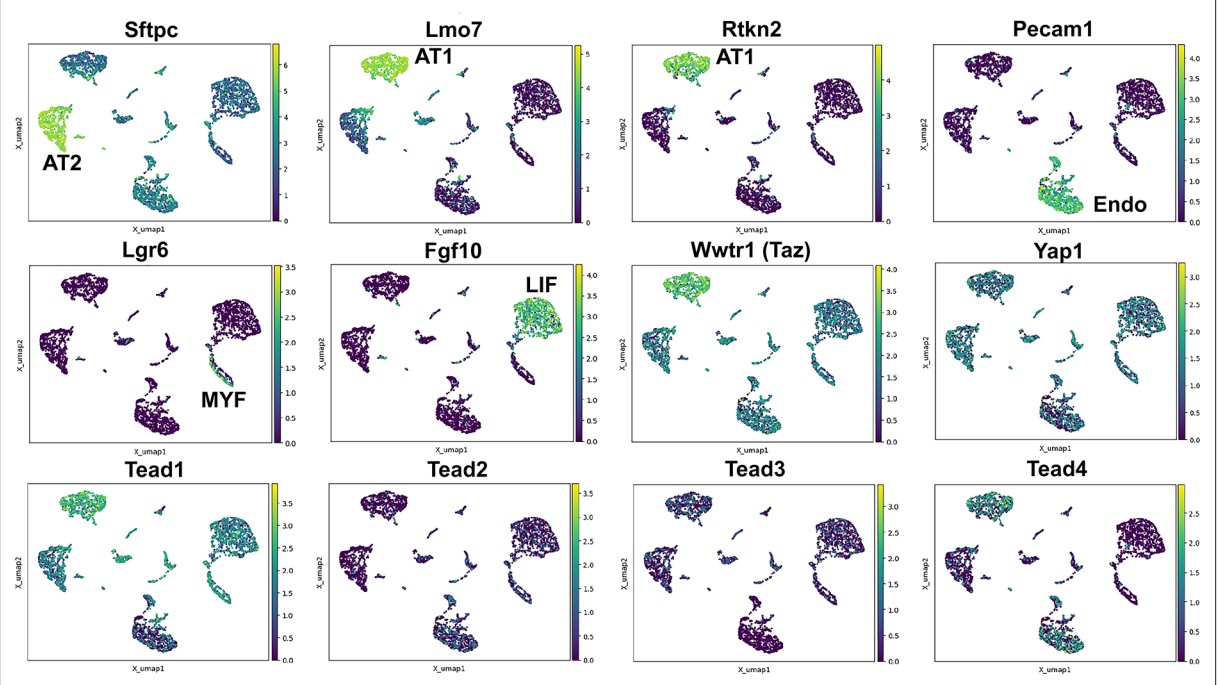

**Figure 1.** Hippo pathway expression in the lung. Single nuclei RNA-seq was performed on E18.5 lungs. (Top and middle row) Cell types were identified based on gene expression: Sftpc (AT2); Lmo7, Rtkn2 (AT1); Pecam1 (endothelial); Lgr6 (myofibroblast/airway smooth muscle); Fgf10 (lipofibroblast). (MIddle and bottom row) Tead 1–4, Wwtr1 (Taz), and Yap1 mapped to these cell types.

## Results

### AT2 cells are actively maintained by Hippo signaling

We have previously shown that in the developing lung heterozygous inactivation of epithelial *Yap1* results in a striking lung phenotype *Volckaert et al., 2019*. We found that, in the lung epithelium, Yap suppresses *Wwtr1* (Taz) expression and that cytoplasmic Yap inhibits β-catenin signaling. We further showed that nuclear Taz is specifically expressed in AT1 cells and that epithelial inactivation of *Wwtr1*, but not *Yap1*, impaired AT1 cell differentiation *Volckaert et al., 2019*.

To better evaluate the expression of Hippo pathway mediators in the lung we performed single nuclear RNAseq on E18.5 ctrl lungs. We chose this time point as most lung cells have been differentiated at this time and the lung is still small enough that isolating nuclei at this stage likely would allow us to sequence all major cell types. Interestingly, we find that *Wwtr1* is expressed the highest in AT1 cells which also express *Tead1* and *Tead4*. However, while *Tead1* is expressed also at lower levels in other lung epithelial cells the expression of *Tead4* was specific to AT1 cells. Together these data further support an important role for Taz-Tead4 mediated transcription in AT1 cell specification and maintenance (*Figure 1*).

Because many developmental pathways (Fgf, Shh, Wnt, etc) are also important in maintaining homeostasis and become activated in the response to injury in the adult lung, we sought to determine how the Hippo pathway maintains homeostasis and promotes regeneration following bleomycin injury in AT2 stem cells. Here, we report that in the adult lung Merlin (*Nf2*), the best-known upstream activator of the Hippo pathway is highly expressed in AT2 but not AT1 cells during homeostasis. We further find strong phosho-Mst1/2 staining in AT2 cells during homeostasis, suggesting that active Hippo signaling, preventing nuclear Yap/Taz localization (*Figure 2A–I*), is required for AT2 stem cell maintenance. Just like during lung development, AT1 cells in the adult lung were characterized by nuclear Taz staining (*Figure 2D–F*), suggesting that inactivation of the Hippo pathway in AT2 cells might be necessary for their differentiation into AT1 cells. Indeed, upon bleomycin injury, we find reduced phosho-Mst1/2 staining in regenerated alveolar epithelial cells (*Figure 2J–L*) compared to uninjured lungs (*Figure 2G–I*). To investigate whether inactivation of the Hippo pathway is sufficient to drive AT2 to AT1 cell differentiation we inactivated *Nf2* or the downstream hippo kinases *Stk4* and

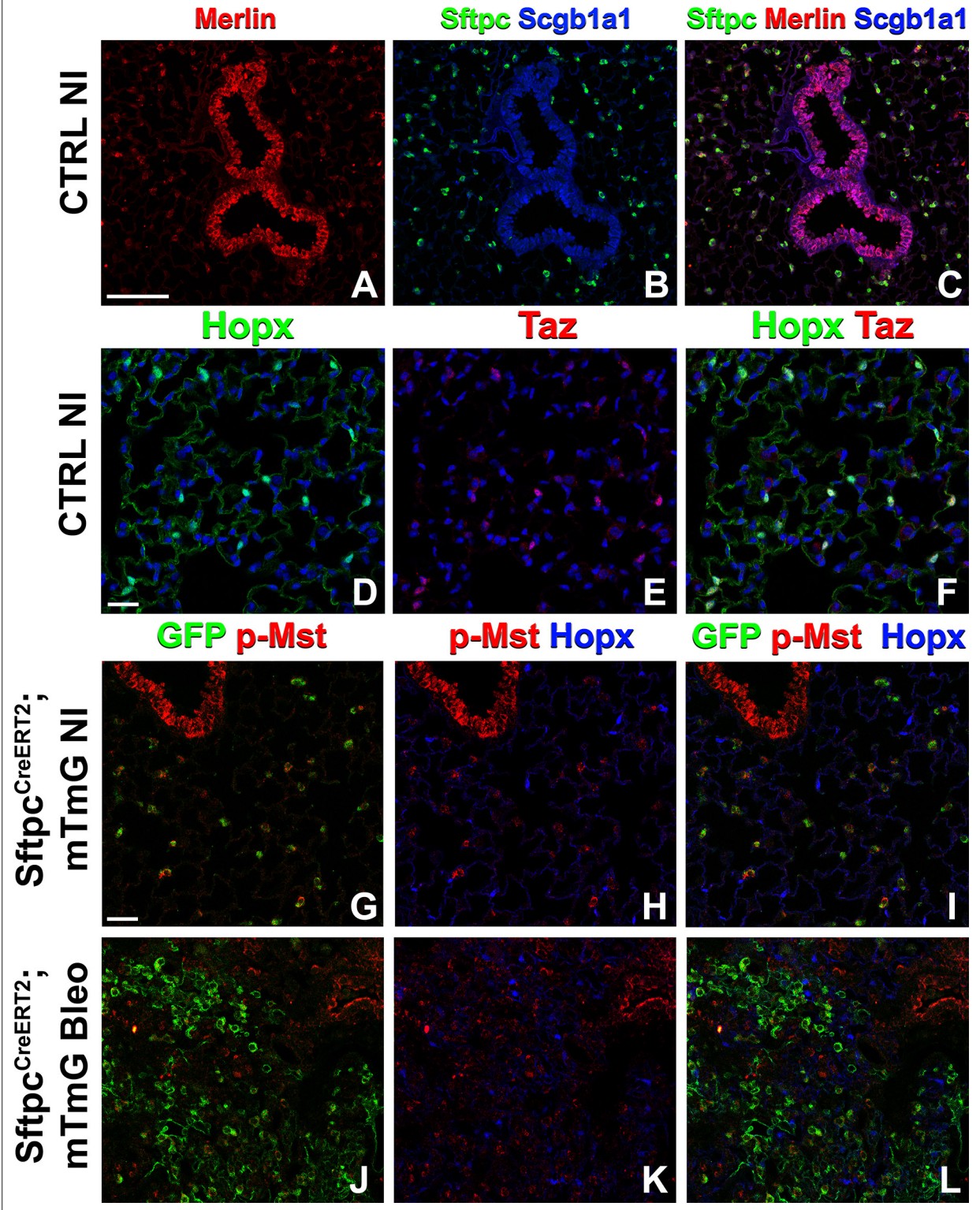

**Figure 2.** The Hippo pathway is active in alveolar type 2 (AT2) cells but not in AT1 cells. Left lung lobes of 8-week-old mice were inflation fixed, embedded in paraffin, and sectioned. (**A–C**) Immunostaining for Sftpc (**A–C**), Merlin (**A, C**), and Scgb1a1 (**B, C**) and (**D–F**) immunostaining for Hopx (**D, F**) and Taz (**E, F**) on control non-injured (NI) lungs. (**G–L**) At 8 weeks of age, mice were placed on tamoxifen chow for 3 weeks, and left lung lobes were inflation fixed, embedded in paraffin, and sectioned 9 weeks after being placed on normal chow (at 20 weeks of age). Intratracheal bleomycin was administered 3 weeks after mice were removed from tamoxifen and placed on normal chow. Bleomycin-injured lungs were harvested at 6 weeks after injury (at 20 weeks of age). Immunostaining for GFP (**G–L**), phosphorylated-Mst (p-Mst; **G, I, J, L**), and Hopx (**H, I, K, L**) on non-injured (NI) *Sftpc^{CreERT2}*;

*Figure 2 continued on next page*

*Figure 2 continued*

*mTmG* (**G–I**) and on bleomycin injured *Sftpc^CreERT2^; mTmG* (**J–L**) lungs. Representative images are presented. Scale bars, 100 µm (**A–C**), 50 µm (**D–F**), 50 µm (**G–L**).

*Stk3* (Mst1 and Mst2, respectively) in adult AT2 cells and found that this drives their spontaneous differentiation and flattening into AT1 cells (***Figure 3A–H–***). However, simultaneous or single inactivation of *Yap1* and *Wwtr1* in AT2 cells did not result in an obvious homeostatic lung phenotype in the absence of injury suggesting that Yap and/or Taz are not required for AT2 stem cell maintenance (***Figure 3I–L and U*** and data not shown).

## Taz promotes alveolar epithelial regeneration and the resolution of bleomycin-induced fibrosis

To investigate the role of epithelial hippo signaling in pulmonary fibrosis, we set out to inactivate different components of the Hippo pathway in AT2 cells prior to bleomycin injury while simultaneously lineage tracing them. To investigate the requirement for Yap and/or Taz in alveolar epithelial regeneration and fibrosis development upon bleomycin injury, we inactivated *Yap1* or *Wwtr1* alone and in combination in AT2 cells. Inactivation of *Wwtr1* alone or in combination with *Yap1* caused a drastic decrease in AT2 to AT1 cell differentiation, increased pulmonary fibrosis (***Figure 4A, B, E–H and K–N*** & ***Figure 4—figure supplement 1***), and increased mortality in the latter (data not shown). However, the inactivation of *Yap1* by itself in AT2 cells resulted in increased AT2 to AT1 cell differentiation and decreased pulmonary fibrosis based on hydroxyproline content and significantly less *Col1a1* and *Col3a1* RNA expression (***Figure 4A–D and K–N*** & ***Figure 4—figure supplement 1***). We found that *Yap1* deficient alveolar epithelial cells showed increased Taz after bleomycin injury (***Figure 4—figure supplement 2***), in line with our previous findings that epithelial inactivation of *Yap1* during lung development results in increased epithelial Taz expression and suggesting that Taz rather than Yap is the critical downstream effector of the Hippo pathway that drives AT1 cell differentiation.

To investigate whether increased nuclear Taz would promote alveolar epithelial regeneration and fibrosis resolution, we performed bleomycin injury on mice in which we inactivated *Nf2* or *Stk3/4* in AT2 cells and found that this resulted in increased AT1 cell regeneration and decreased pulmonary fibrosis based on hydroxyproline content (***Figure 4A, B, I, J and K–O*** & ***Figure 4—figure supplement 1***). *Sftpc^CreERT2^; Nf2^f/f^* mice also featured decreased *Col1a1* and *Col3a1* expression (***Figure 4O***).

## Impaired alveolar epithelial regeneration upon inactivation of Yap/Taz in AT2 cells results in increased bronchiolization

We performed spatial transcriptomics on Control (Ctrl), *Sftpc^CreERT2^; Stk3/4^f/f^, and Sftpc^CreERT2^; Yap1^f/f^, Wwtr1^f/f^* lungs in the absence of injury or 6 weeks post bleomycin injury for lung epithelial cell markers and found that as expected compare to control lungs AT1 markers (*Rtkn2, Cav1, Cldn18, Vegfa*) were strongly reduced in *Sftpc^CreERT2^; Yap1^f/f^, Wwtr1^f/f^* lungs before and after bleomycin injury (***Figure 5***). Compared to control lungs, the AT2 marker (*Lamp3*) was reduced in *Sftpc^CreERT2^; Stk3/4^f/f^* lungs in the absence of injury in line with our immunostaining findings that *Stk3/4* inactivation in AT2 cells causes their premature differentiation into AT1 cells. Additionally, *Lamp3* was reduced in *Sftpc^CreERT2^; Yap1^f/f^, Wwtr1^f/f^* lungs after injury coinciding with a decrease of cells entering the S phase in *Sftpc^CreERT2^; Yap1^f/f^, Wwtr1^f/f^* lungs after bleomycin injury (***Figure 5—figure supplement 1***). We also found an increase in AT1 markers (*Rtkn2, Cav1, Cldn18, Vegfa*) in *Sftpc^CreERT2^; Stk3/4^f/f^* lungs compared to control lungs in the absence of injury.

However, unexpectedly we found increased bronchiolization (*Scgb3a2, Ltf, Krt5, Krt15, Trp63*) and an apparent increase in basal pods (*Krt5, Krt15, Trp63*) in bleomycin injured *Sftpc^CreERT2^; Yap1^f/f^, Wwtr1^f/f^* lungs compared to bleomycin injured Ctrl or *Sftpc^CreERT2^; Stk3/4^f/f^* lungs (***Figure 6***). Bronchiolization is a hallmark of IPF but is only commonly observed upon severe or repetitive bleomycin injury in mice (***Figure 6—figure supplement 1***) ***Redente et al., 2021***. Immunostaining and qPCR for *Krt5* and *Trp63* confirmed that *Sftpc^CreERT2^; Yap1^f/f^, Wwtr1^f/f^*, and to a lesser extent *Sftpc^CreERT2^; Wwtr1^f/f^* lungs showed a drastic increase in basal cell pods after bleomycin injury (***Figure 7A–C***). In addition, *Sftpc^CreERT2^; Yap1^f/f^, Wwtr1^f/f^* lungs featured a drastic increase in *Muc5b* RNA expression (***Figure 7A–C***). Together, our findings indicate that impaired alveolar epithelial regeneration and/or increased alveolar injury upon

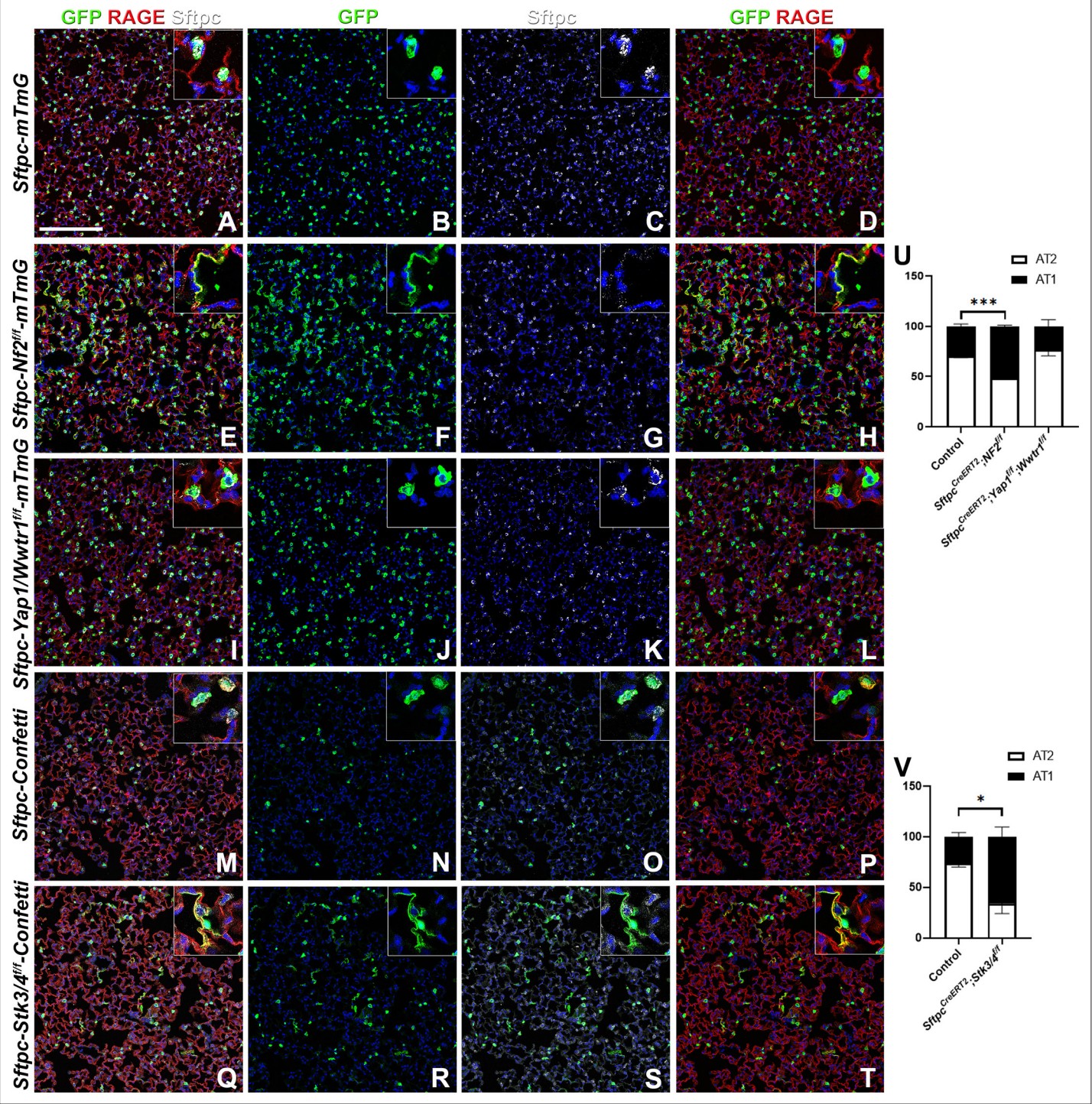

**Figure 3.** The Hippo pathway actively maintains alveolar type 2 (AT2) cells during homeostasis. At 8 weeks of age, mice were placed on tamoxifen chow for 3 weeks, and left lung lobes were inflation fixed, embedded in paraffin, and sectioned 9 weeks after being placed on normal chow (at 20 weeks of age). (**A–T**) Immunostaining for GFP (**B, F, J, N, R**), Sftpc (**C, G, K, O, S**), and RAGE (**D, H, L, P, T**) on *Sftpc$^{CreERT2}$; mTmG* (A-D, n=4) *Sftpc$^{CreERT2}$; Nf2$^{f/f}$mTmG* (E-H, n=3), *Sftpc$^{CreERT2}$; Yap1$^{f/f}$; Wwtr1$^{f/f}$; mTmG* (I-L, n=4), *Sftpc$^{CreERT2}$; Confetti* (M-P, n=3), *Sftpc$^{CreERT2}$; Stk3/4$^{f/f}$;Confetti* (**Q-T**, n=3). Representative images are presented. Scale bar, 200 μm. (**U**) Image analysis GFP⁺/Sftpc⁺ (AT2) and GFP⁺/RAGE⁺ (AT1) on staining in A-L and (**V**) M-T. Data are mean ± SEM. Student's t-test was used to determine significance. *p<0.05, **p<0.01, ****p<0.0001.

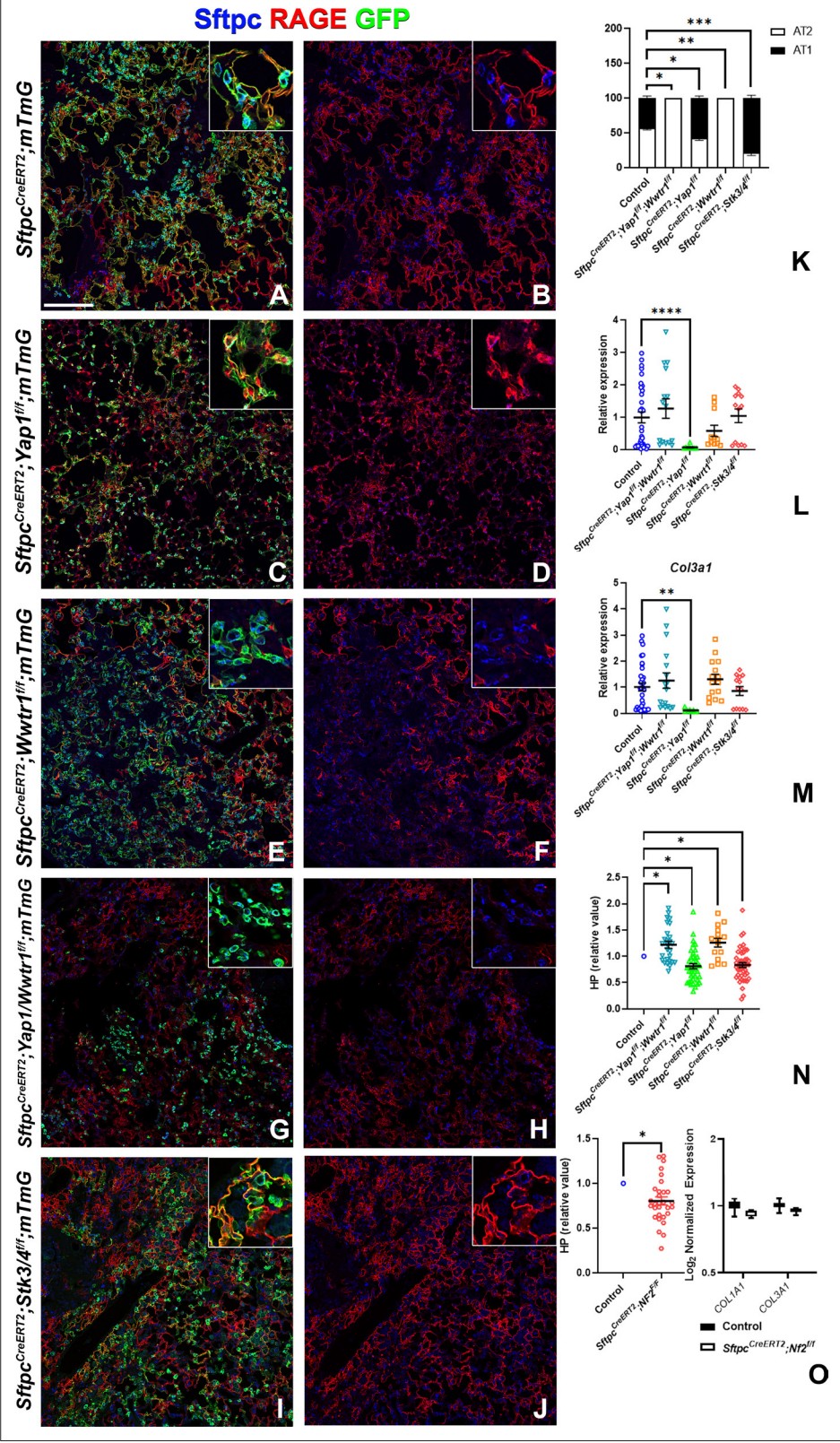

**Figure 4.** Taz is required for alveolar type 2 (AT2) into AT1 cell differentiation following bleomycin injury. At 8 weeks of age, mice were placed on tamoxifen chow for 3 weeks, and following a 3-week washout period, mice were injured with intratracheal administration of bleomycin. At 6 weeks post-injury (at 20 weeks of age), left lung lobes were inflation fixed, embedded in paraffin, and sectioned. (A–J) Immunostaining for GFP, Sftpc, and RAGE

*Figure 4 continued on next page*

*Figure 4 continued*

on *Sftpc^CreERT2*; *mTmG* (A, B, n=3), *Sftpc^CreERT2*; *Yap1^f/f*;*mTmG* (C, D, n=4), *Sftpc^CreERT2*; *Wwtr1^f/f*; *mTmG* (E, F, n=5), *Sftpc^CreERT2*;*Yap1^f/f*; *Wwtr1^f/f*; *mTmG* (G, H, n=4), *Sftpc^CreERT2*; *Stk3/4^f/f*; *mTmG* (I, J, f/f=6). Representative images are presented. Scale bar, 200 μm. (**K**) Image analysis GFP⁺/Sftpc⁺ (AT2) and GFP⁺/RAGE⁺ (AT1) on staining in A-J. (**L–M**) qPCR analysis for *Col1a1* (**L**) and *Col3a1* (**M**) on control (n=33), *Sftpc^CreERT2*; *Yap1^f/f*; *Wwtr1^f/f* (n=15), *Sftpc^CreERT2*; *Yap1^f/f* (n=9), *Sftpc^CreERT2*; *Wwtr1^f/f* (n=11), *Sftpc^CreERT2*; *Stk3/4* f/f (n=13). Values are represented as $2^{-\Delta\Delta Ct}$ normalized to Control. (**N**) Hydroxyproline analysis for soluble collagen on control, *Sftpc^CreERT2*; *Yap1^f/f*; *Wwtr1^f/f* (n=27, control n=25), *Sftpc^CreERT2*; *Yap1^f/f* (n=40, control n=37), *Sftpc^CreERT2*; *Wwtr1^f/f* (n=15, control n=10), *Sftpc^CreERT2*; *Stk3/4* f/f (n=46, control n=53). Values are normalized to each genotype's Cre- control. (**O**) Hydroxyproline analysis for soluble collagen on control (n=15) and *Sftpc^CreERT2*; *Nf2^f/f* (n=30) and $\text{Log}_2$ normalized values for RNA expression for *Col1a1* and *Col3a1* from NanoString analysis (control n=21, *Sftpc^CreERT2*; *Nf2^f/f* n=6). Values are normalized to control. (**K-N**) Data are mean ± SEM. (O) Data are mean ± SEM (left panel) or box and whisker plot (right panel). Student's t-test was used to determine significance. *p<0.05, **p<0.01, ****p<0.0001.

The online version of this article includes the following figure supplement(s) for figure 4:

**Figure supplement 1.** Inactivation of *Stk3/4* promotes fibrosis resolution 6 weeks after bleomycin injury.

**Figure supplement 2.** Inactivation of *Yap1* in alveolar type 2 (AT2) induces Taz upregulation.

inactivation of *Yap1/Wwtr1* in AT2 cells results in increased bronchiolization. This finding is very interesting and suggests that alveolar epithelial and bronchial epithelial stem cells compete against each other to re-epithelialize the injured lung after injury. Bronchiolization in *Sftpc^CreERT2*; *Yap1^f/f*, *Wwtr1^f/f* lungs after bleomycin injury resembled the bronchiolization observed after catastrophic H1N1 influenza-mediated lung injury *Kumar et al., 2011*; *Ray et al., 2016*.

## Activation of profibrotic transcriptional programs upon inactivation of Yap/Taz in AT2 cells

We next analyzed predicted transcription factor activity in our spatial transcriptomics data and found increased *Trp63* activity in *Sftpc^CreERT2*; *Yap1^f/f*, *Wwtr1^f/f* lungs consistent with the increase in basal cells. Differential analysis of transcription factor activity in our different mutant lungs indicated that one of the most strikingly differentially activated transcription factors was the profibrotic transcription factor *Sp3 Lee et al., 2020*; *Kum et al., 2007*; *Wohlfahrt et al., 2019*, the activity of which is upregulated in the bronchiolarized areas of bleomycin injured *Sftpc^CreERT2*; *Yap1^f/f*, *Wwtr1^f/f* lungs (*Figure 7—figure supplement 1*, *Figure 7—figure supplement 2*). While the role of Sp3 in the lung is unclear, Sp3-deficient embryos are growth retarded and invariably die at the birth of respiratory failure. We further found an upregulation in Nr1h3 (Lxra) and SRF activity in *Sftpc^CreERT2*; *Yap1^f/f*, *Wwtr1^f/f* lung after injury. Nr1h3 (Lxra) is an activator of SREBP-1 which suppresses Srebf1-activation of fibroblasts and the progression of pulmonary fibrosis *Shichino et al., 2019*. Serum response factor (SRF) on the other hand has been shown to be essential for myofibroblast differentiation and the development of pulmonary fibrosis *Bernau et al., 2021*. Together, these data suggest that the lack of AT1 cell differentiation leads to increased pro-fibrotic transcription factor activity and promotes fibrosis at the expense of epithelial regeneration in bleomycin-injured lungs.

## Discussion

In summary, our data suggest that facilitating AT1 cell differentiation after injury promotes the resolution of pulmonary fibrosis. Our data indicate that increased nuclear Taz drives alveolar epithelial regeneration and fibrosis resolution, since both loss of *Yap1* or inactivation of *Stk3/4* or *Nf2,* which all result in increased nuclear Taz promote alveolar epithelial regeneration and fibrosis resolution, whereas inactivation of *Wwtr1* alone impairs regeneration and fibrosis resolution. As such our findings clearly demonstrate that Yap and Taz play distinctive, even contrasting roles in AT2 stem cells in driving alveolar epithelial regeneration upon bleomycin injury.

While our data is partially supported by previous works, we also demonstrate a possible injury-specific regulation by Taz. In a *S. pneumoniae* infection injury, simultaneous inactivation of both *Yap1* and *Wwtr1* resulted in impaired alveolar epithelial regeneration. However, single *Yap1* or *Wwtr1* mutant mice had minor or no phenotypes *LaCanna et al., 2019*. We provide evidence that after bleomycin injury, single inactivation of *Yap1* promotes alveolar regeneration by increasing nuclear

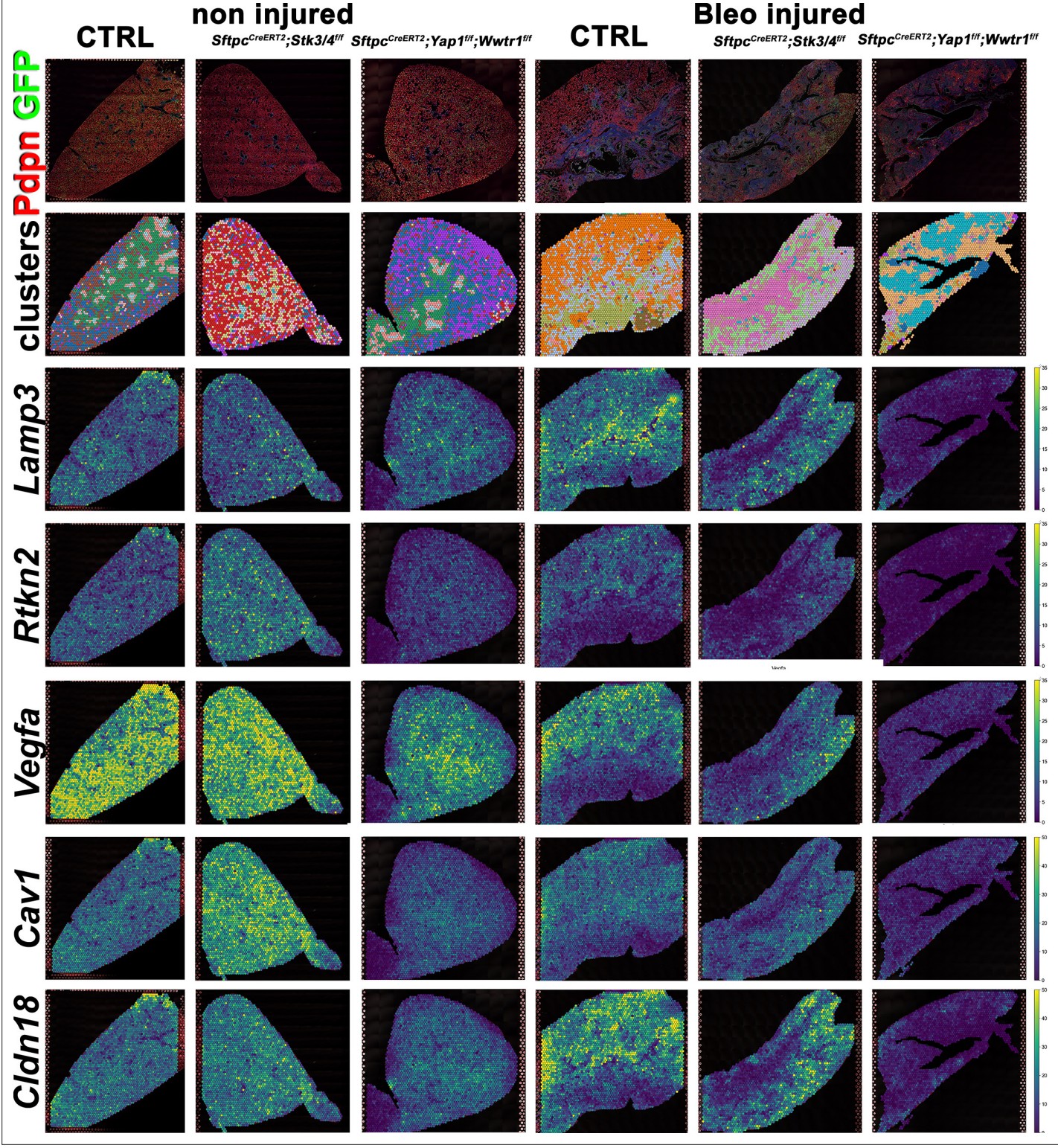

**Figure 5.** Decreased alveolar epithelial regeneration following bleomycin injury upon inactivation of *Yap/Taz* in alveolar type 2 (AT2) cells. At 8 weeks of age, mice were placed on tamoxifen chow for 3 weeks, and following a 3-week washout period, mice were non-injured or injured with intratracheal administration of bleomycin. At 6 weeks post-injury (at 20 weeks of age), left lung lobes were inflation fixed, embedded in paraffin, and sectioned. Spatial transcriptomics was performed on injured control *Sftpc^CreERT2^; mTmG, Sftpc^CreERT2^; Yap1^f/f^;Wwtr1^f/f^, mTmG, Sftpc^CreERT2^; Stk3/4^f/f^; mTmG* and non-injured control *Sftpc^CreERT2^; mTmG, Sftpc^CreERT2^; Yap1^f/f^; Wwtr1^f/f^, mTmG, Sftpc^CreERT2^; Stk3/f/f^f/f^; mTmG* lungs. Immunofluorescence colocalization of Pdpn (red) and GFP (green) on non-injured lungs and lungs 6 weeks after bleomycin injury. Projection of spot clusters onto immunofluorescence image of

*Figure 5 continued on next page*

*Figure 5 continued*

the tissue sample. Spatial gene expression transcripts of cell type-specific markers were mapped onto spot coordinates. *Lamp3* (AT2) and *Rtkn2, Vegfa, Cav1, Cldn18* (AT1) markers. Color scale reflects the abundance of indicated transcripts.

The online version of this article includes the following figure supplement(s) for figure 5:

**Figure supplement 1.** Alveolar type 2 (AT2) inactivation of *Yap1/Wwtr1* reduces the S phase after bleomycin injury.

Taz in AT2 cells. Additionally, in a pneumonectomy model, complete *Yap1* inactivation in AT2 cells reduced their proliferation and differentiation compared to heterozygous inactivation of *Yap* in AT2 cells *Liu et al., 2016*. These findings raise the intriguing possibility that reducing Yap levels rather than complete inactivation of Yap has additional advantages in regard to promoting alveolar epithelial regeneration.

Activation of Yap and Taz has been demonstrated in fibrosis of other organs (reviewed in *Kim et al., 2019* and *Mia and Singh, 2022*). In cardiac fibrosis, heart failure patients had increased nuclear Yap and increased proliferation of cardiac fibroblasts in left ventricular heart tissue *Sharifi-Sanjani et al., 2021*. In a mouse model of myocardial infarction, knocking out *Yap1* and *Wwtr1* or *Yap1* alone in cardiac fibroblasts, reduced fibrosis after infarction *Mia et al., 2022*; *Francisco et al., 2020*. Interestingly, inactivation of *Lats1/2*, which results in increased nuclear Yap/Taz, in cardiomyocytes, reduced fibrosis after infarction suggesting a cell-specific response to infarction *Shao et al., 2021*. Additionally, in the liver, Yap/Taz was elevated in liver hepatocytes patients with fibrotic nonalcoholic steatohepatitis (NASH). Inactivation of *Yap1* and *Wwtr1* or *Wwtr1* alone in hepatocytes reduced liver fibrosis in a mouse model of NASH *Wang et al., 2016*; *Mooring et al., 2020*. In patients with liver fibrosis, hepatic stellate cells (HSCs), the primary source of fibroblasts in liver fibrosis, feature elevated nuclear Yap compared to healthy control livers *Mannaerts et al., 2015*. Inhibiting the Yap-Tead binding with verteporfin prevented fibrosis development in a mouse model of liver fibrosis by inhibiting HSC activation and collagen deposition *Mannaerts et al., 2015*; *Martin et al., 2016*; *Yu et al., 2019*. In renal fibrosis, elevated Yap was identified in renal fibroblasts after unilateral ureteral obstruction (UUO), a mouse model of renal fibrosis, and inactivation of *Yap1/Wwtr1* reduced the expression of fibronectin, collagen, and α-SMA *Liang et al., 2017*. Together, these data suggest that Yap and Taz are not identical twins, and understanding how each is regulated in a cell type-dependent manner in response to injury is critical to finding a therapy that can alleviate fibrosis in several organ systems.

Interestingly, while many reports suggest that Yap and Taz are equivalent and redundant, our data suggest that these Hippo effectors have nonredundant functions in terms of AT1 cell differentiation. We provide evidence that in response to bleomycin-induced pulmonary fibrosis, increased Taz through inactivation of *Yap1, Stk3/4,* or *Nf2* is vital for AT1 cell differentiation and reduction of fibrosis. Interestingly, we found that in the absence of AT1 regeneration in *Sftpc^{CreERT2}; Wwtr1^{f/f}* or *Sftpc^{CreERT2}; Yap1^{f/f}; Wwtr1^{f/f}*, there is an increase in bronchiolization, a hallmark of IPF, and increase in pro-fibrotic transcription factor activity in the latter. Thus, promoting Taz activity in alveolar stem cells may promote fibrosis resolution by inducing AT1 cell differentiation.

## Materials and methods
### Experimental model and subject details
All mice were bred and maintained in a pathogen-free environment with free access to food and water. Both male and female mice were used for all experiments. *Sftpc^{CreERT2} Chapman et al., 2011*, *Rosa26-mTmG* (JAX 007676; RRID:IMSR_JAX:007676), *Rosa26-Confetti* (JAX 017492; RRID:IMSR_JAX:017492), *Stk3/4^{f/f}* (JAX 017635; RRID:IMSR_JAX:017635), *Yap1^{f/f} Xin et al., 2013*, *Wwtr1^{f/f 44}*, *Nf2^{f/f} Giovannini et al., 2000* mice were previously described.

For bleomycin injury, adult 8- to 12-week-old mice were intratracheally instilled with 50 uL bleomycin (0.8–2 U/kg body weight optimized for each strain, batch of bleomycin, and gender) as described previously *Redente et al., 2014*. For tamoxifen induction, mice were placed on tamoxifen-containing food (rodent diet with 400 mg/kg tamoxifen citrate; Harlan Teklad TD.130860) for 3 weeks and *Sftpc-CreERT2* mice received two additional intraperitoneal tamoxifen injections (0.20 mg/g body weight, Enzo Life Sciences) in the last week of tamoxifen citrate feed. All experiments were approved by the Mayo Clinic Institutional Animal Care and Use Committee.

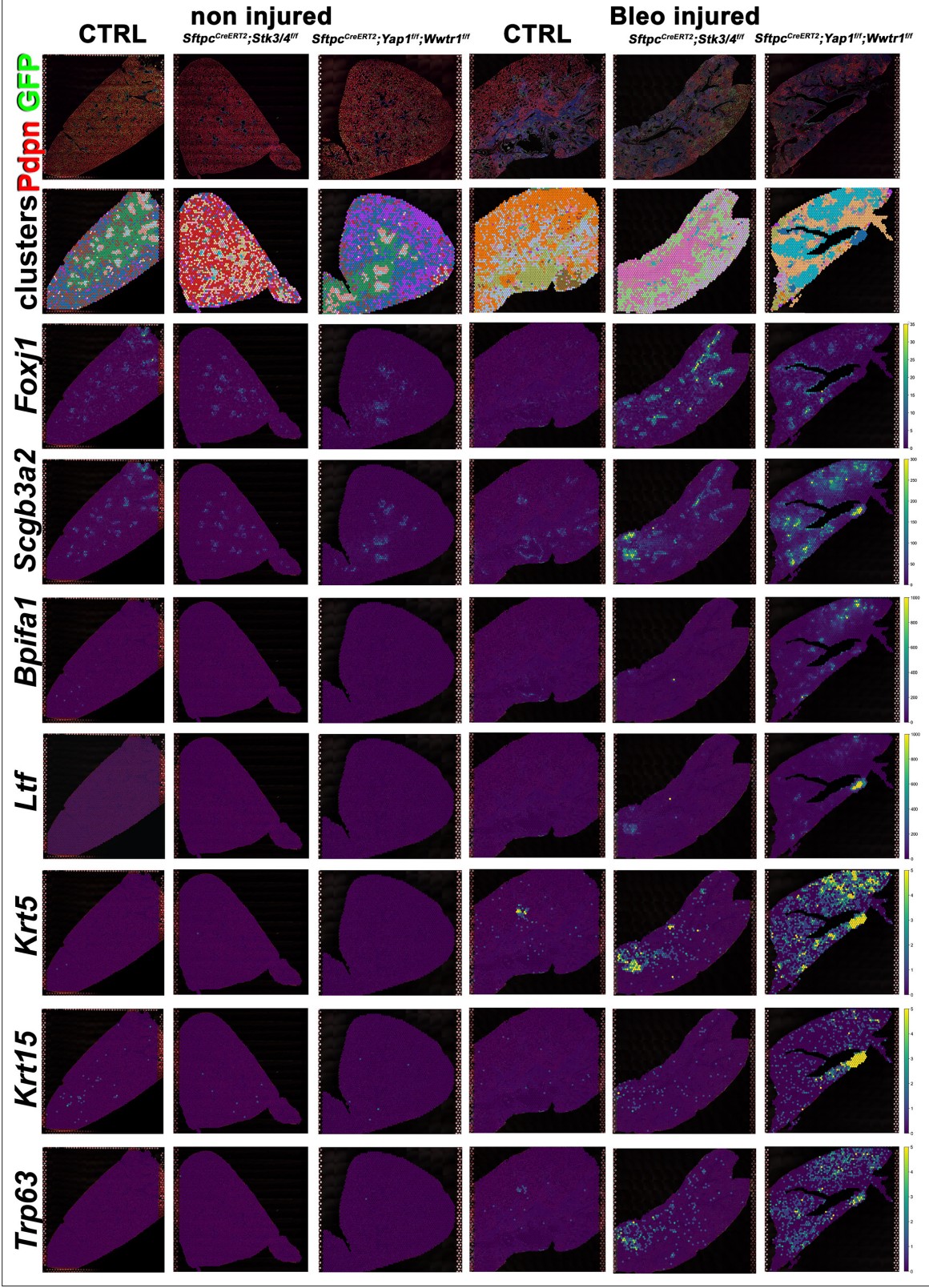

**Figure 6.** Increased bronchiolization upon *Yap/Taz* inactivation in alveolar type 2 (AT2) cells. At 8 weeks of age, mice were placed on tamoxifen chow for 3 weeks, and following a 3-week washout period, mice were non-injured or injured with intratracheal administration of bleomycin. At 6 weeks post-injury (at 20 weeks of age), left lung lobes were inflation fixed, embedded in paraffin, and sectioned. Spatial transcriptomics was performed on injured (6 weeks after bleomycin injury) control *Sftpc^CreERT2^; mTmG, Sftpc^CreERT2^; Yap1^f/f^;Wwtr1^f/f^, mTmG, Sftpc^CreERT2^; Stk3/4^f/f^; mTmG* and non-injured control

*Figure 6 continued on next page*

*Figure 6 continued*

*Sftpc^CreERT2^; mTmG, Sftpc^CreERT2^; Yap1^f/f^; Wwtr1^f/f^, mTmG, Sftpc^CreERT2^; Stk3/ f/f^f/f^; mTmG* lungs. Immunofluorescence colocalization of Pdpn (red) and GFP (green) on non-injured lungs and lungs 6 weeks after bleomycin injury. Projection of spot clusters onto immunofluorescence image of the tissue sample. Spatial gene expression transcripts of cell type-specific markers were mapped onto spot coordinates. *Foxj1* (ciliated), *Scgb3a2* (serous and club), *Bpifa1, Ltf* (serous) and *Krt5, Krt15, Trp63* (basal) markers. Color scale reflects the abundance of indicated transcripts.

The online version of this article includes the following figure supplement(s) for figure 6:

**Figure supplement 1.** Repetitive bleomycin doses induce bronchiolization.

## Immunohistochemistry and fluorescence

All staining was done on paraffin sections of formalin-fixed lungs or tracheas. Immunofluorescent staining was performed with the following primary antibodies: rabbit anti-Merlin (NF2; 1:250; clone A-19; sc-331; RRID:AB_2298548; Santa Cruz Biotechnology), rabbit anti-WWTR1 (Taz; 1:250; clone E115179; HPA007415; RRID:AB_1080602; Prestige Antibodies), rabbit anti-phosporylated-Mst1(Thr183)/2(Thr180) (1:200; 3681; RRID:AB_330269; Cell Signaling), goat anti-Scgb1a1 (1:200;

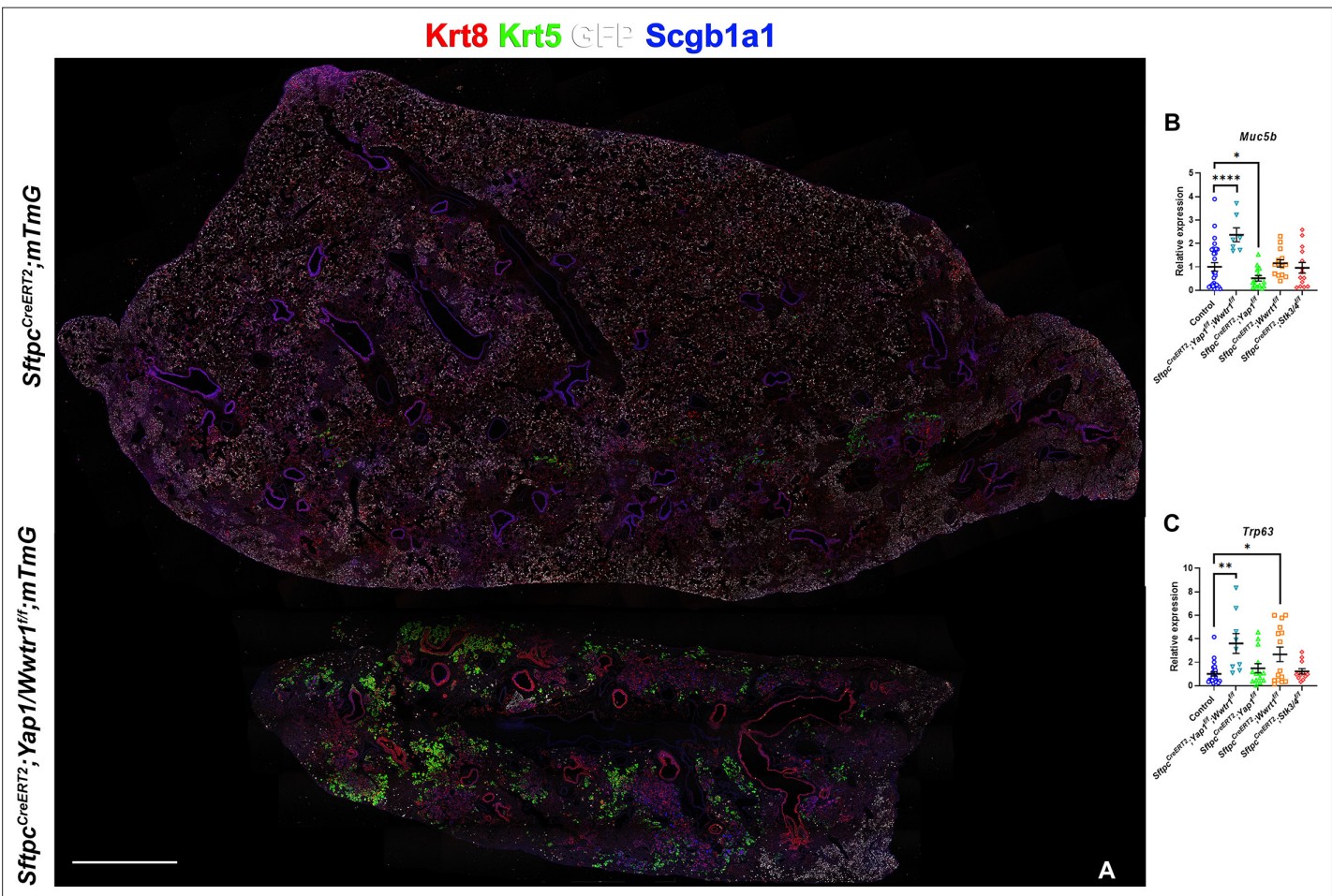

**Figure 7.** Inactivation of Yap and Taz in alveolar type 2 (AT2) cells enhances bronchiolization following bleomycin injury. At 8 weeks of age, mice were placed on tamoxifen chow for 3 weeks, and following a 3-week washout period, mice were injured with intratracheal administration of bleomycin. At 6 weeks post-injury (at 20 weeks of age), left lung lobes were inflation fixed, embedded in paraffin, and sectioned. (**A**) Immunostaining for GFP, Krt8, Krt5, and Scgb1a1 on control *Sftpc^CreERT2^; mTmG* and *Sftpc^CreERT2^; Yap1^f/f^; Wwtr1^f/f^, mTmG*. Scale bar, 666 µm. (**B–C**) qPCR analysis for *Muc5b* (**B**) and *Trp63* (**I**), on control (n=29), *Sftpc^CreERT2^; Yap1^f/f^; Wwtr1^f/f^* (n=7), *Sftpc^CreERT2^; Yap1^f/f^* (n=14), *Sftpc^CreERT2^; Wwtr1^f/f^* (n=14), *Sftpc^CreERT2^; Stk3/4^f/f^* (n=14) lungs. Data are mean ± SEM. Student's t-test was used to determine significance. *p<0.05, **p<0.01, ****p<0.0001.

The online version of this article includes the following figure supplement(s) for figure 7:

**Figure supplement 1.** Predicted transcription factor expression.

**Figure supplement 2.** Predicted transcription factor expression is altered based on genotype and injury status.

clone T-18; sc-9772; RRID:AB_2238819; Santa Cruz Biotechnology Inc), chicken anti-GFP (1:250; GFP-1020; RRID:AB_10000240; Aves Labs Inc), rabbit anti-Keratin 5 (1:200; clone EP1601Y; MA5-14473; RRID:AB_10979451; Thermo Fisher Scientific), rabbit anti-SFTPC (1:200; WRAB-9337; RRID:AB_2335890; Seven hills bioreagents), rat anti-RAGE (1:500; Clone 175410; MAB1179; RRID:AB_2289349; R&D Systems), mouse anti-HOP (1:100; Clone E-1; sc-398703; RRID:AB_2687966; Santa Cruz Biotechnology Inc), rat anti-keratin 8 (1:100; TROMA-I; RRID:AB_531826; Developmental Studies Hybridoma Bank), and Syrian hamster anti-podoplanin (PDPN, T1a; 1:500; 8.1.1; RRID:AB_531893; Developmental Studies Hybridoma Bank). After deparaffinization, slides were rehydrated through a series of decreasing ethanol concentrations, antigen unmasked by either microwaving in citrate-based antigen unmasking solution (Vector Labs, H-3000) or by incubating sections with proteinase K (7.5 µg/ml) (Invitrogen, 25530–049) for 7 min at 37 °C. Tissue sections were then washed in TBS with 0.1% Tween-20 and blocked with 3% Bovine Serum Albumin (BSA), and 0.4% Triton in TBS for 30 min at room temperature followed by overnight incubation of primary antibodies diluted in 3% BSA, 0.1% Triton in TBS. The next day, slides were washed in TBS with 0.1% Tween-20 and incubated with secondary antibodies diluted in 3% BSA, and 0.1% Triton in TBS for 3 hr at room temperature. All fluorescent staining was performed with appropriate secondary antibodies from Jackson Immunoresearch. Slides were mounted using Vectashield (Vector Labs, H-1000). For alcian blue staining, the slides were deparaffinized and hydrated with distilled water, stained in a 1% Alcian blue solution (pH 2.5) for 30 min. After washing them in water for 2 min, the slides were dehydrated with ethanol, cleared in xylene, and mounted with a mounting medium.

## Microscopy and imaging

Tissue was imaged using a micrometer slide calibrated Zeiss LSM800 Laser scanning confocal microscope using ZEN imaging software or Leica Stellaris 5 confocal microscope with LASX imaging software. Lungs were imaged using tiled stitched 20 x images covering the entire cross-section of the left or lower right lung lobe from ≥6 different lungs. Representative images were chosen. Images were processed and analyzed using Zen blue (Zeiss) or LASX and Adobe Photoshop Creative Suite 3 (Adobe) software.

## Image quantification

Differentiation of GFP-positive cells was determined using artificial intelligence and machine learning image segmentation with Zeiss Zen Blue Intellesis or ImageJ software. The total area of GFP and GFP overlapping with different cell-specific antibody stains (Sftpc or RAGE) was determined. Image quantification and analysis were performed in a double-blinded fashion. Each quantification was ≥3 different lung images.

## Quantitative real-time PCR

Total mRNA was extracted from lung accessory lobes stored in RNAlater (Invitrogen, AM7021) and using Total RNA Kit I (Omega Biotek, R6834-02) according to the manufacturer's instructions. RNA concentration was determined by spectrophotometry. cDNA was generated using Maxima First Strand cDNA Synthesis (Fisher Scientific, FERK1642) according to the manufacturer's instructions. Gene expression was analyzed by quantitative RT-PCR using TaqMan Gene Expression Assays (Applied Biosystems, 4369016) directed against the mouse targets *β-glucuronidase* (Mm00446953_m1), *Trp63* (Mm00495788_m1), *Muc5b* (Mm00466391_m1), *Col1a1* (Mm00801666_g1), *Col3a1* (Mm01254476_m1). Quantitative real-time PCR was performed using a StepOne Plus system (Applied Biosystems). Data were presented as $2^{-\Delta\Delta Ct}$ with *β-glucuronidase* as the internal sample control normalized to the control group. Each experiment was repeated with samples obtained from ≥6 different lung preparations.

## Nanostring

RNA was isolated from lung accessory lobes as described above. 100 ng of RNA was hybridized with a custom RNA probe panel designed by NanoString (NanoString Technologies) for 16 hr according to the manufacturer's instructions. The RNA-probe hybridization was loaded on a NanoString cartridge and processed in a NanoString nCounter. Data were analyzed with Rosalind.bio (Rosalind, Inc) and

Log2 Fold Changes were calculated and graphed. Each experiment was repeated with samples obtained from ≥3 different lung preparations.

## Single nuclei RNA-sequencing

Embryonic day 18.5 (E18.5) lungs were collected from WT mice and flash-frozen in a dry ice/ethanol slurry. Frozen E18.5 tissues were homogenized with a dounce homogenizer and nuclei were isolated according to 10 x Genomics Nuclei Isolation from Embryonic Mouse Brain for Single Cell Multiome ATAC+ Gene Expression Sequencing. Nuclei viability was assessed and total nuclei were counted. ATAC and gene expression libraries were then constructed from the nuclei according to the 10 x Genomics Chromium Next GEM Single Cell Multiome ATAC+ Gene Expression User Guide. Sequencing was performed on a NextSeq 500. Alignment was performed using the 'Cellranger count' function provided in 10 x Genomics single-cell gene expression software.

### Secondary analysis

The Python packages Scanpy and Muon were used to apply standard quality control metrics and unsupervised clustering for the generation of initial UMAP projections. Muon was used to integrate multiomics data.

## Spatial transcriptomics

RNA was isolated from formalin-fixed paraffin-embedded (FFPE) tissue sections using E.Z.N.A FFPE RNA Kit (Omega Bio-Tek). The RNA integrity in FFPE blocks was determined on an Agilent TapeStation. 5 µm FFPE lung sections that had a DV200% above 50 were placed within the frames of the capture areas on the active surface of the Visium spatial slide (10 x Genomics) and processed according to the manufacturer's instructions. Tissues were stained with podoplanin (PDPN, T1a) and GFP and imaged with fluorescent secondary antibodies. Final library preparations and sequencing were completed by the Mayo Genomics Research Core according to the manufacturer's instructions on an Illumina NextSeq. Count matrices were generated using the 'spaceranger count' function in Space Ranger 1.0.0. The resulting data were processed in Scanpy, Squidpy, and Decoupler. The Decoupler DoRothEA wrapper was used to predict transcription factor activity. DoRothEA is a comprehensive resource containing a curated collection of transcription factors (TFs) and their target genes.

## Hydroxyproline

The right lobes were flash-frozen in dry ice at the time of harvest and stored at –80 °C. For acid hydrolysis, the lobes were baked in a 70 °C oven without lids for 2 days until completely dry. The weights of dry lobes were measured and 500 µl of 6 N HCl was added to each sample. The lungs were then hydrolyzed in an 85 °C oven for 2 days with occasional vortexing. The hydrolysates were cooled at room temperature and centrifuged at maximum speed for 10 min. The supernatants then were transferred to fresh 1.5 mL tubes and centrifuged at maximum speed for 10 min. Each sample or standard was diluted with citrate-acetate buffer (5% citric acid, 1.2% glacial acetic acid, 7.24% sodium acetate, and 3.4% sodium hydroxide) in a 96-well plate. Chloramine-T solution (1.4% chloramine-T, 10% N-propanol, and 80% citrate-acetate buffer) was added, and the mixture was incubated for 20 min at room temperature. Then, Ehrlich's solution (1.27 M p-dimethylaminobenzaldehyde, 70% N-propanol, 20% perchloric acid) was added to each sample and the samples were incubated at 65 °C for 20 min. Absorbance was measured at 550 nm. Standard curves were generated for each experiment using the reagent hydroxyproline (Sigma H-1637) as a standard. The amount (µg) of hydroxyproline was calculated by comparison to the standard curve.

## Quantification and statistical analysis

All results are expressed as mean values ± SEM. The 'n' represents biological replicates and can be found in the figure legends. The significance of differences between two sample means was determined by unpaired two-tailed student's t-test (assuming unequal or equal variances as determined by the F-test of equality of variances). All datasets followed a normal distribution and p-values less than 0.05 were considered statistically significant. The number of samples to be used was based on the number of experimental paradigms multiplied by the number in each group that is necessary to yield

statistically significant results based on power analysis, to reject the null hypothesis with 80% power (type I error = 0.05).

## Acknowledgements

This study was supported by NIH R01 HL146461, HL132156, and NIH R35 HL161169 to SDL. RW was supported by NIH T32 HL105355.

## Additional information

### Funding

| Funder | Grant reference number | Author |
|---|---|---|
| National Heart, Lung, and Blood Institute | R35 HL161169 | Stijn P De Langhe |
| National Heart, Lung, and Blood Institute | R01 HL146461 | Stijn P De Langhe |
| National Heart, Lung, and Blood Institute | R01 HL132156 | Stijn P De Langhe |
| National Heart, Lung, and Blood Institute | T32 HL105355 | Rachel Warren |

The funders had no role in study design, data collection and interpretation, or the decision to submit the work for publication.

### Author contributions

Rachel Warren, Resources, Data curation, Formal analysis, Validation, Investigation, Visualization, Methodology, Writing – original draft, Writing – review and editing; Handeng Lyu, Data curation, Formal analysis, Investigation, Methodology; Kylie Klinkhammer, Investigation, Methodology; Stijn P De Langhe, Conceptualization, Resources, Data curation, Software, Formal analysis, Supervision, Funding acquisition, Validation, Investigation, Visualization, Methodology, Writing – original draft, Project administration, Writing – review and editing

### Author ORCIDs

Rachel Warren ⓘ http://orcid.org/0000-0001-9438-2975
Stijn P De Langhe ⓘ http://orcid.org/0000-0003-3867-4572

### Ethics

This study was performed in strict accordance with the recommendations in the Guide for the Care and Use of Laboratory Animals of the National Institutes of Health. All of the animals were handled according to approved institutional animal care and use committee (IACUC) protocols (#A00006342-21) of the Mayo clinic.

### Decision letter and Author response

Decision letter https://doi.org/10.7554/eLife.85092.sa1
Author response https://doi.org/10.7554/eLife.85092.sa2

## Additional files

### Supplementary files
• MDAR checklist

### Data availability
Sequencing data have been uploaded to Dryad.

The following dataset was generated:

| Author(s) | Year | Dataset title | Dataset URL | Database and Identifier |
|---|---|---|---|---|
| Warren R, Lyu H, Gao S, Klinkhammer K, De Langhe SP | 2022 | Hippo signaling impairs alveolar epithelial regeneration in pulmonary fibrosis | https://doi.org/10.5061/dryad.tdz08kq3d | Dryad Digital Repository, 10.5061/dryad.tdz08kq3d |

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
