## [Editor Report]

This is an interesting and potentially significant study that adds important new information to our understanding of the mechanisms of lung epithelial repair after tissue injury. The authors have delineated a novel and non-redundant role for the hippo pathway and the downstream regulators Yap/Taz in regulating the repair of lung injury. These studies will inform future investigations into the mechanisms of the repair of lung injury.

---

## [Decision Letter]

**Decision letter after peer review:**

Thank you for submitting your article "Hippo signaling impairs alveolar epithelial regeneration in pulmonary fibrosis" for consideration by *eLife*. Your article has been reviewed by 3 peer reviewers, and the evaluation has been overseen by a Reviewing Editor and Paul Noble as the Senior Editor. The reviewers have opted to remain anonymous.

Essential revisions:

All three reviewers found significant merit in the study. Please address the comments provided by the reviewers:

1. It would be helpful to include data points on graphs in figures 3, 4 and 7, and associated "n" in figure legends.

2. Many of the immunofluorescence images would benefit from inclusion of high magnification insets for emphasis.

3. The extent of Merlin immunoreactivity within AT2 cells (Figure 2) would be more clearly demonstrated if individual channels were shown for Merlin/Sftpc colocalization in addition to a merged image.

This interesting paper could be strengthened by a careful temporal dissection of the role of YAP and TAZ, since these effectors may have different functional roles in early vs. late phases of lung repair and regeneration. Currently, all models are based on genetic disruption prior to injury with tamoxifen induction starting 3 weeks before bleomycin injury. An important experiment to substantiate the conclusion that epithelial TAZ is protective is to perform "gain of function" studies in established fibrosis to determine whether this reverses fibrosis or accelerates resolution.

Methodological approaches rely heavily on image analyses, and more quantitative assessments should be included; for example, in Figure 4, the inclusion of hydroxyproline measurements are shown "relative" to the bleomycin-injured wild-type mice, but lacks the un-injured control group, thus it is difficult to assess the relative effects in genetically altered mice (note that "n" values are not shown, and a dot plot figure rather than mean with SD/SEM would be preferred).

Please present a rationale for the choice of a 6 week post bleo time point in terms of its significance in terms of progressive fibrosis versus repair and regeneration. Last sentence page 9 "promotes fibrosis..." at the expense of what?

The transition from embryonic to adult lung studies could be made more smoothly and highlight the significance in a bit more detail. page 6.

The discussion is brief and a bit redundant with the Introduction. The authors may be better served to discuss their current findings in detail with regard to the existing literature (Hippo signaling components in lung and other organs).

Figures:

Please include detail of age at harvest in the Figure legends (it is present in some).

Please include color intensity scale in figures 5and6 and sup Figures3and5.

The immunostaining is elegant but zoomed panels may be appropriate to include for visualization.

Figure 2 AT 1 cells in the adult were characterized by nuclear Taz (page 6) however it doesn't look like 100%. Perhaps quantitate and state if/why there may be heterogeniety and discuss the significance.

Figure 4 A/B may illustrate some loss of tissue structure (large empty spaces) in addition to the appearance of fibrosis. Is this the case or is the observed space the choice of representative image?

*Reviewer #1 (Recommendations for the authors):*

Some concerns with data presentation include:

1. It would be helpful to include data points on graphs in figures 3, 4 and 7, and associated "n" in figure legends.

2. Many of the immunofluorescence images would benefit from inclusion of high magnification insets for emphasis.

3. The extent of Merlin immunoreactivity within AT2 cells (Figure 2) would be more clearly demonstrated if individual channels were shown for Merlin/Sftpc colocalization in addition to a merged image.

*Reviewer #2 (Recommendations for the authors):*

This interesting paper could be strengthened by a careful temporal dissection of the role of YAP and TAZ, since these effectors may have different functional roles in early vs. late phases of lung repair and regeneration. Currently, all models are based on genetic disruption prior to injury with tamoxifen induction starting 3 weeks before bleomycin injury. An important experiment to substantiate the conclusion that epithelial TAZ is protective is to perform "gain of function" studies in established fibrosis to determine whether this reverses fibrosis or accelerates resolution.

Methodological approaches rely heavily on image analyses, and more quantitative assessments should be included; for example, in Figure 4, the inclusion of hydroxyproline measurements are shown "relative" to the bleomycin-injured wild-type mice, but lacks the un-injured control group, thus it is difficult to assess the relative effects in genetically altered mice (note that "n" values are not shown, and a dot plot figure rather than mean with SD/SEM would be preferred).

*Reviewer #3 (Recommendations for the authors):*

Comments to the authors:

The manuscript Hippo signaling impairs alveolar epithelial regeneration in pulmonary fibrosis is a very rigorous and timely report of the significance of Taz and Yap in AT2/AT1 differentiation and their impact on the progression of fibrosis versus repair and tissue regeneration. The manuscript is very well written and the detail in the Introduction was appreciated. This reviewer is requesting additional detail to improve overall conceptualization of the data.

lease present a rationale for the choice of a 6 week post bleo time point in terms of its significance in terms of progressive fibrosis versus repair and regeneration. Last sentence page 9 "promotes fibrosis…" at the expense of what?

The transition from embryonic to adult lung studies could be made more smoothly and highlight the significance in a bit more detail. page 6.

The discussion is brief and a bit redundant with the Introduction. The authors may be better served to discuss their current findings in detail with regard to the existing literature (Hippo signaling components in lung and other organs).

Figures:

Please include detail of age at harvest in the Figure legends (it is present in some).

Please include color intensity scale in figures 5and6 and sup Figures3 and 5.

The immunostaining is elegant but zoomed panels may be appropriate to include for visualization.

Figure 2 AT 1 cells in the adult were characterized by nuclear Taz (page 6) however it doesn’t look like 100%. Perhaps quantitate and state if/why there may be heterogeniety and discuss the significance.

Figure 4 A/B may illustrate some loss of tissue structure (large empty spaces) in addition to the appearance of fibrosis. Is this the case or is the observed space the choice of representative image?

---

## [Author Response]

Essential revisions:All three reviewers found significant merit in the study. Please address the comments provided by the reviewers:1. It would be helpful to include data points on graphs in figures 3, 4 and 7, and associated "n" in figure legends.

We have now included data points for the graphs in which it was possible and in included the associated “n” for all graphs in the figure legends.

2. Many of the immunofluorescence images would benefit from inclusion of high magnification insets for emphasis.

We have now included high magnification insets for most immunofluorescence images.

3. The extent of Merlin immunoreactivity within AT2 cells (Figure 2) would be more clearly demonstrated if individual channels were shown for Merlin/Sftpc colocalization in addition to a merged image.

We have now altered Figure 2 to better demonstrate Merlin staining in AT2 cells.

This interesting paper could be strengthened by a careful temporal dissection of the role of YAP and TAZ, since these effectors may have different functional roles in early vs. late phases of lung repair and regeneration. Currently, all models are based on genetic disruption prior to injury with tamoxifen induction starting 3 weeks before bleomycin injury. An important experiment to substantiate the conclusion that epithelial TAZ is protective is to perform "gain of function" studies in established fibrosis to determine whether this reverses fibrosis or accelerates resolution.

The mouse models used in this manuscript do not allow us to specifically target AT2 cells after injury. As Sftcp^CreERT2^ will activate in other epithelial cell types during regeneration. For this reason, we do feel it is appropriate to inactivate *Mst_1/2_* or *Nf2* in Sftpc^+^ cells post injury.

We did harvest lungs at 3 weeks after bleomycin injury for *Sftpc^CreERT2^;Yap^f/f^;Taz^f/f^* and *Sftpc^CreERT2^;Mst_1/2_^f/f^* mice and have now included HP data for this time point in the manuscript. In addition, we have now included HP data on non-injured mice for each genotype.

Methodological approaches rely heavily on image analyses, and more quantitative assessments should be included; for example, in Figure 4, the inclusion of hydroxyproline measurements are shown "relative" to the bleomycin-injured wild-type mice, but lacks the un-injured control group, thus it is difficult to assess the relative effects in genetically altered mice (note that "n" values are not shown, and a dot plot figure rather than mean with SD/SEM would be preferred).

We have now changed the figures to dot plots and have included non-injured data in new Figure S1.

Please present a rationale for the choice of a 6 week post bleo time point in terms of its significance in terms of progressive fibrosis versus repair and regeneration. Last sentence page 9 "promotes fibrosis….." at the expense of what?

Since pulmonary fibrosis in the bleomycin mouse model tends to mostly resolve by 6-8 weeks, we focused on the 6 week time point. As such we think our data better reflects whether our mice develop progressive fibrosis and decreased regeneration as is the case for *Sftpc^CreERT2^;Yap^f/f^;Taz^f/f^* or increased regeneration and resolution of pulmonary fibrosis as is the case for *Sftpc^CreERT2^;Mst_1/2_^f/f^* mice. We have now also included new 3 week HP time points for these 2 strains Figure S1. To better illustrate that regeneration is reduced vs enhanced in *Sftpc^CreERT2^;Yap^f/f^;Taz^f/f^* and *Sftpc^CreERT2^;Mst_1/2_^f/f^* mice respectively. Not that even *Sftpc^CreERT2^;Yap^f/f^;Taz^f/f^* mice show reduced fibrosis at 6 weeks compared to 3 weeks which is likely because airway epithelial cells can contribute to regeneration somewhat.

The transition from embryonic to adult lung studies could be made more smoothly and highlight the significance in a bit more detail. page 6.

We have added a transition here.

The discussion is brief and a bit redundant with the Introduction. The authors may be better served to discuss their current findings in detail with regard to the existing literature (Hippo signaling components in lung and other organs).

We have discussed literature regarding fibrosis in other organs and the role of the Hippo pathway in those organs.

Figures:Please include detail of age at harvest in the Figure legends (it is present in some).

We have now included this information.

Please include color intensity scale in figures 5and6 and sup Figures3and5.

We have now included the color intensity scale for these figures.

The immunostaining is elegant but zoomed panels may be appropriate to include for visualization.

We have now included high magnification insets for most immunofluorescence images.

Figure 2 AT 1 cells in the adult were characterized by nuclear Taz (page 6) however it doesn't look like 100%. Perhaps quantitate and state if/why there may be heterogeniety and discuss the significance.

The images are confocal so if nuclear Taz appears lower in some cells it’s likely due to the particular confocal plane of this image.

Figure 4 A/B may illustrate some loss of tissue structure (large empty spaces) in addition to the appearance of fibrosis. Is this the case or is the observed space the choice of representative image?

These lungs are less fibrotic and therefore probably better inflated but it partially due to the choice or image.

Reviewer #1 (Recommendations for the authors):Some concerns with data presentation include:1. It would be helpful to include data points on graphs in figures 3, 4 and 7, and associated "n" in figure legends.2. Many of the immunofluorescence images would benefit from inclusion of high magnification insets for emphasis.3. The extent of Merlin immunoreactivity within AT2 cells (Figure 2) would be more clearly demonstrated if individual channels were shown for Merlin/Sftpc colocalization in addition to a merged image.

These points have been addressed above.

Reviewer #2 (Recommendations for the authors):This interesting paper could be strengthened by a careful temporal dissection of the role of YAP and TAZ, since these effectors may have different functional roles in early vs. late phases of lung repair and regeneration. Currently, all models are based on genetic disruption prior to injury with tamoxifen induction starting 3 weeks before bleomycin injury. An important experiment to substantiate the conclusion that epithelial TAZ is protective is to perform "gain of function" studies in established fibrosis to determine whether this reverses fibrosis or accelerates resolution.Methodological approaches rely heavily on image analyses, and more quantitative assessments should be included; for example, in Figure 4, the inclusion of hydroxyproline measurements are shown "relative" to the bleomycin-injured wild-type mice, but lacks the un-injured control group, thus it is difficult to assess the relative effects in genetically altered mice (note that "n" values are not shown, and a dot plot figure rather than mean with SD/SEM would be preferred).

These points have been addressed above.

Reviewer #3 (Recommendations for the authors):Comments to the authors:The manuscript Hippo signaling impairs alveolar epithelial regeneration in pulmonary fibrosis is a very rigorous and timely report of the significance of Taz and Yap in AT2/AT1 differentiation and their impact on the progression of fibrosis versus repair and tissue regeneration. The manuscript is very well written and the detail in the Introduction was appreciated. This reviewer is requesting additional detail to improve overall conceptualization of the data.lease present a rationale for the choice of a 6 week post bleo time point in terms of its significance in terms of progressive fibrosis versus repair and regeneration. Last sentence page 9 "promotes fibrosis…" at the expense of what?The transition from embryonic to adult lung studies could be made more smoothly and highlight the significance in a bit more detail. page 6.

We have added a transition here.

The discussion is brief and a bit redundant with the Introduction. The authors may be better served to discuss their current findings in detail with regard to the existing literature (Hippo signaling components in lung and other organs).

We have added information regarding fibrosis in other organs and the role of the Hippo pathway in those organs.

Figures:Please include detail of age at harvest in the Figure legends (it is present in some).Please include color intensity scale in figures 5and6 and sup Figures3 and 5.The immunostaining is elegant but zoomed panels may be appropriate to include for visualization.Figure 2 AT 1 cells in the adult were characterized by nuclear Taz (page 6) however it doesn’t look like 100%. Perhaps quantitate and state if/why there may be heterogeniety and discuss the significance.Figure 4 A/B may illustrate some loss of tissue structure (large empty spaces) in addition to the appearance of fibrosis. Is this the case or is the observed space the choice of representative image?

These points have been addressed above.